# Therapeutic Approaches to Targeting Androgen Receptor Splice Variants

**DOI:** 10.3390/cells13010104

**Published:** 2024-01-04

**Authors:** Violet A. Daniels, Jun Luo, Channing J. Paller, Mayuko Kanayama

**Affiliations:** 1Department of Urology, James Buchanan Brady Urological Institute, Johns Hopkins University School of Medicine, Baltimore, MD 21287, USA; vdaniel5@jhmi.edu (V.A.D.); jluo2@jhmi.edu (J.L.); 2Departments of Oncology, Sidney Kimmel Comprehensive Cancer Center, Johns Hopkins University School of Medicine, Baltimore, MD 21287, USA

**Keywords:** androgen receptor splice variants, AR-V7, castration-resistant prostate cancer

## Abstract

Therapeutic options for advanced prostate cancer have vastly expanded over the last decade and will continue to expand in the future. Drugs targeting the androgen receptor (AR) signaling pathway, i.e., androgen receptor targeting agents (ARTAs), remain the mainstream treatments that are increasingly transforming the disease into one that can be controlled for an extended period of time. Prostate cancer is inherently addicted to AR. Under the treatment pressure of ARTA, molecular alterations occur, leading to the clonal expansion of resistant cells in a disease state broadly categorized as castration-resistant prostate cancer (CRPC). One castration resistance mechanism involves AR splice variants (AR-Vs) lacking the ligand-binding domain. Some AR-Vs have been identified as constitutively active, capable of activating AR signaling pathways without androgenic ligands. Among these variants, AR-V7 is the most extensively studied and may be measured non-invasively using validated circulating tumor cell (CTC) tests. In the context of the evolving prostate cancer treatment landscape, novel agents are developed and evaluated for their efficacy in targeting AR-V7. In patients with metastatic CRPC (mCRPC), the availability of the AR-V7 tests will make it possible to determine whether the treatments are effective for CTC AR-V7-positive disease, even though the treatments may not be specifically designed to target AR-V7. In this review, we will first outline the current prostate cancer treatment landscape, followed by an in-depth review of relatively newer prostate cancer therapeutics, focusing on AR-targeting agents under clinical development. These drugs are categorized from the standpoint of their activities against AR-V7 through direct or indirect mechanisms.

## 1. Introduction

Prostate cancer is the second-leading cause of cancer-related deaths among men in the United States [1]. This warrants a large body of research to decipher the intricate biology of the disease and devise effective treatments. The hallmark of prostate cancer progression is the androgen receptor (AR) signaling pathway, which is mainly responsible for prostate cell survival, proliferation, and resistance to treatment. Despite significant advancements in the targeting of this pathway, the emergence of castration-resistant prostate cancer (CRPC) and its inherent resistance to conventional anti-androgen therapies have underscored the complexity of the disease. Metastatic CRPC (mCRPC), under pressure from regimens targeting the AR pathway, can undergo molecular changes driving resistance to these treatment regimens [2]. A pivotal facet of this resistance arises from alternative splicing of AR mRNA, generating AR splice variants (AR-Vs) [3]. AR-Vs are truncated isoforms of the AR that do not bind the androgens but have the potential to be constitutively active. Among the AR-Vs, AR-V7 is the most researched splice variant. AR-V7 lacks the ligand binding domain (LBD), as shown in Figure 1, which allows it to activate the AR signaling pathway in the absence of androgens [4]. Suppression of ligand-mediated AR-FL signaling results in increased AR-V7 expression in CRPC cell line models [5]. AR-V7 activates a distinct expression signature enriched for cell cycle genes without requiring the presence of androgens or functional AR-FL [5], though AR-FL continues to be expressed in higher abundance [4,5,6]. This shift toward AR-V7-mediated signaling after suppression of ligand-dependent AR-FL is considered an important mechanism contributing to drug resistance to CRPC therapy [5]. 

In mCRPC patients, positive circulating tumor cell (CTC) AR-V7 is associated with progressive diseases resistant to AR-targeting agents [7,8]. However, it is important to note that AR-V7 may not be the stand-alone driver of castration resistance [3]. Regardless, when present, these variants challenge the therapeutic paradigm by indicating therapy resistance and treatment failure. Consequently, targeting AR-Vs has emerged as an imperative focus of research and development [9].

This review aims to provide a comprehensive, up-to-date overview of the diverse therapeutic approaches to counter the influence of AR-Vs as a follow-up to our last review [3]. It delves into the existing strategies that directly and indirectly target AR-Vs, exploring the novel compounds poised to revolutionize the precision of AR-V targeted therapy, and shedding light on ongoing clinical trials that hold promise to transform these strategies into tangible benefits for patients. In the relentless pursuit of improved prostate cancer management, understanding and targeting AR-Vs, given the availability of a non-invasive blood-based AR-V7 test, is a necessary step toward addressing therapeutic resistance and improving patient outcomes. Considering the substantial effort underway to develop drugs designed to target AR-V7, we will focus on agents undergoing clinical development. Since clinical development of these novel agents will occur in the ever-changing treatment landscape, this review will first provide an overview of the current prostate cancer treatment landscape, followed by a comprehensive literature review of drugs that either directly or indirectly target AR-Vs.

**Figure 1 cells-13-00104-f001:**
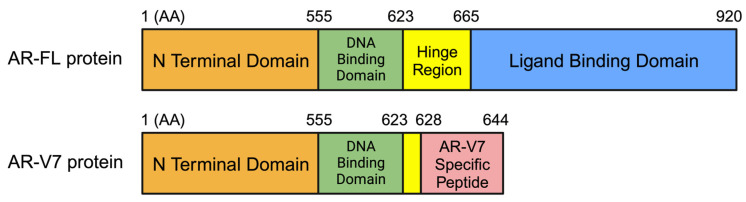
AR-FL vs. AR-V7 Protein Functional Domains. AR-V7 lacks most of the hinge region and the entire ligand binding domain present in AR-FL. The figure is not drawn to scale. Source: Adapted from Figure 1 of the Luo 2016 review [10]. Numbers indicate amino acid (AA) numbers. Protein length slightly varies among literatures due to the variation in CAG repeat numbers in NTD. The amino acid position annotations in this figure are based on NP_000035.2 for AR-FL and NP_001334990.1 for AR-V7. AR-V7 protein possesses a 16-AA AR-V7-specific peptide at its C-terminal end.

## 2. Prostate Cancer Treatment Landscape

Traditional therapeutic strategies for treating prostate cancer involve a sequential progression through available treatment modalities in response to evolving disease status. Upon the initial diagnosis of localized prostate cancer, often detected through early screening, interventions encompass local treatments, including surgery and radiation. About one-third of patients with localized disease experience recurrence following local therapies [11]. Disease recurrence is often indicated by prostate-specific antigen (PSA) (i.e., biochemical recurrence) or imaging. Those encountering a biochemical or local recurrence might continue under observation, bypassing ADT to mitigate its potential side effects, until reaching a predetermined threshold, subsequently leading to the adoption of ADT in the form of luteinizing hormone-releasing hormone (LHRH) antagonists or agonists. This regimen may persist, with intermittent periods of treatment holidays, until signs of disease progression via PSA, imaging, or clinical symptoms become apparent [12]. While these initial treatments can yield remission for patients—terminology that is handled cautiously in prostate cancer oncology—most individuals will eventually develop diseases resistant to ADT, that is, CRPC.

In cases where the malignancy has extended beyond its original site before local treatments, the healthcare provider and the patient need to make a judicious decision about the initiation of androgen-deprivation therapy (ADT) either as a singular agent, in combination with radiation for low-volume metastatic disease [13], or with docetaxel for both low- and high-volume metastatic disease [14]. 

In instances where a patient receives an initial diagnosis of metastatic prostate cancer (de novo mHSPC), the standard approach involves the prompt initiation of ADT with anti-androgen therapies such as bicalutamide, abiraterone, or enzalutamide plus or minus docetaxel [15]. Over time, patients may exhibit signs of progression as their tumor tissue develops resistance to AR-targeted therapies. Consequently, alternative therapeutic avenues, including poly (ADP-ribose) polymerase (PARP) inhibitors, chemotherapy, radium 223, radiation therapy, prostate-specific membrane antigen (PSMA)-targeted treatments, and immunotherapy, are explored [12]. Figure 2 shows this progression through treatments. 

## 3. Drugs That Target AR-Vs Directly

Table 1 summarizes agents having putative roles in suppressing AR-Vs through direct functional inhibition, including niclosamide, TAS3681, EPI-7386, Au-AR pep-PROTAC, VNPP433-3β, and RIPTAC. We provide a detailed review of these agents currently undergoing clinical development and summarize their putative mechanisms of action in Figure 3. 

### 3.1. Niclosamide

Within the landscape of existing therapies and compounds undergoing clinical trials, several avenues hold promise for directly targeting AR-Vs. A notable contender is niclosamide, which exhibits potent inhibitory effects on AR-V7. Niclosamide’s mechanism of action for prostate cancer is expected to involve hindering AR-V7 recruitment to androgen-responsive elements such as the PSA promoter and curbing its protein expression by instigating degradation [16]. Furthermore, preclinical data suggest this compound demonstrates proficiency in curbing prostate cancer cell proliferation in vitro and mitigating tumor growth in vivo [16]. Notably, niclosamide not only overcomes resistance to enzalutamide but also synergizes with enzalutamide therapy, thus portraying its potential utility in advanced prostate cancer cases, particularly those resistant to enzalutamide [16]. 

The therapeutic trajectory of niclosamide encountered hurdles in 2018 during a Phase I dose-escalation study testing oral niclosamide plus standard-dose enzalutamide in men with mCRPC previously treated with abiraterone (NCT02532114). The trial revealed that the doses needed to attain efficacious plasma concentrations yielded substantial toxicity, precluding the viability of niclosamide as an oral anticancer agent. Consequently, the trial was prematurely halted due to futility, casting uncertainty over the future development of niclosamide as a cancer therapy due to its poor pharmacokinetic properties [17]. 

Fortunately, further research has been conducted to establish the maximum tolerable dose and recommended dosage of niclosamide. A Phase Ib trial investigating reformulated niclosamide in tandem with abiraterone acetate and prednisone in men with CRPC elicited findings that underscored the combination’s favorable tolerability profile and the achievement of therapeutic niclosamide concentrations (NCT02807805). Encouragingly, select patients recorded noteworthy declines in PSA levels, coupled with radiographic ameliorations. The findings suggested that the combination of niclosamide and abiraterone is clinically active [18]. Phase II of this trial is ongoing.

A recent investigation has endeavored to amend niclosamide’s poor pharmaceutical properties by analyzing the efficacy of a novel series of niclosamide analogs and further characterizing the structure-activity relationship of the compound. Several analogs exhibited equivalent or improved anti-proliferation effects in cell lines, potent AR-V7 downregulation, and improved metabolic activity [19]. Further investigation into the clinical use of these compounds may yield exciting results for the field.

### 3.2. TAS3681

TAS3681 is a promising experimental agent that may have dual capabilities to act against androgens and downregulate AR-FL and AR-Vs. In in vitro and in vivo experiments, TAS3681 has shown effectiveness in prostate cancer cell lines, including those with both normal and alternatively spliced AR, such as AR-V7 [20]. Before entering clinical trials, preclinical research revealed that TAS3681 hinders the movement of AR into the cell nucleus after binding and lowers the levels of AR-FL and AR-V7 [21]. Its potential to combat AR-V7-positive cancer lines through AR downregulation prompted the initiation of a Phase I study for patients with mCRPC (NCT02566772). The ongoing trial investigates the safety, tolerability, and efficacy of the compound through a dose-escalation phase followed by an expansion phase that enrolled patients who progressed on either abiraterone or enzalutamide +/− taxane-based chemotherapy. This study is ongoing, but some preliminary results from the dose-escalation phase published in 2021 demonstrated that TAS3681 has a well-tolerated safety profile and exhibits anticancer effects in patients with mCRPC at 300 mg twice daily [22]. The results from this trial are sure to yield intriguing insights into the practicality of TAS3681 as a compound with the potential to inhibit both AR-FL and AR-V7, though it was designed to target AR-FL.

### 3.3. EPI Compounds

Considering the indispensable role of the androgen-receptor N-terminal domain (AR-NTD), which contains the most transcriptional activity of AR across androgen levels, this domain becomes an attractive target for novel therapeutic interventions. EPI compounds have been shown to inhibit AR transcriptional activity, degrade AR-FL and AR-Vs, and prevent AR-V-associated resistance [23]. In 2010, investigators identified EPI-001, a small molecule that blocked the transactivation of AR-NTD. It interacted with the AF-1 region of AR-NTD, disrupting protein-protein interactions and reducing the binding of ARs to androgen-response elements on target genes. Importantly, EPI-001 effectively blocked androgen-induced cancer cell proliferation and caused a reduction in CRPC tumor growth without causing any noticeable toxicity in mouse models [24]. EPI-001 is a mixture of four stereoisomers, with the most potent stereoisomer being EPI-002. EPI-001 and EPI analogs covalently bound to the NTD, effectively blocking AR and its splice variant’s (ARv567es) transcriptional activity, and showed reduced CRPC tumor growth in experiments [25]. 

EPI-506, a prodrug of EPI-002, emerged as a potential therapeutic approach targeting both full-length AR and AR splice variants responsible for resistance to prevailing treatments [26]. However, a Phase I/II investigation involving men with mCRPC was terminated due to a lack of efficacy and poor bioavailability (NCT02606123). Investigation into another EPI compound, EPI-7386, demonstrated improved pharmaceutical properties [27]. It is now in Phase I/II testing as a monotherapy (NCT04421222) and in combination with enzalutamide (NCT05075577) [28].

### 3.4. PROTAC

A new drug called proteolysis-targeting chimera (PROTAC) has been designed to degrade AR-FL and AR-V7 in a specific manner using LBD, or the DNA binding domain (DBD), and an MDM2-dependent mechanism. The PROTAC agents that target AR-LBD, such as ARV-110 [29], are considered ineffective against AR-Vs since the variants lack that domain. The PROTAC molecules that target the DBD, however, may be effective against AR-Vs. To improve the effectiveness of PROTAC, the drug is delivered using an ultra-small gold-peptide complex platform. The resulting drug, Au-AR pep-PROTAC, efficiently degrades AR-FL and AR-V7 in prostate cancer cells, leading to tumor regression in both sensitive and resistant mouse models [30]. Clinical investigations into this compound have yet to be initiated. 

### 3.5. VNPP433-3β

VNPP433-3β is a galeterone derivative recently developed to concurrently target AR and MNK1/2. Despite its successful progression in Phase 2 clinical trials for the treatment of mCRPC, galeterone encountered challenges of early trial termination due to high censorship for the primary endpoint during the pivotal Phase 3 clinical trial (NCT02438007) [31]. As a result, researchers have developed a galeterone-derivative drug candidate, VNPP433-3β, effective against CRPC by targeting AR-FL and AR-Vs. VNPP433-3β promotes the degradation of full-length AR and AR-V7 while also inhibiting MNK1/2, leading to reduced phosphorylation of eIF4E and mRNA translation. Preclinical investigation with cell lines (CWR22Rv1 and LNCaP) on this novel drug compound has shown promising results [32]. 

### 3.6. RIPTAC

Regulated Induced Proximity Targeting Chimera (RIPTAC) is another novel compound currently being investigated to target ARs. RIPTACs are compounds designed to selectively bind a protein that is overproduced in cancer cells. In the case of prostate cancer, the ideal candidate is AR, which is overexpressed as castration resistance occurs. After the compound binds AR, its mechanism comes into play. It is then able to selectively bind and therefore sequester an effector protein (EP) essential to cell function and viability. The cell dies off as a result if enough of the EP is incorporated in the AR:RIPTAC:EP stable ternary complex [33]. There must be significantly elevated levels of AR in the cell in order for enough essential proteins to be sequestered for the cell to die off. Because of this, RIPTAC-mediated disruption of EP should only affect cancerous cells that are producing AR at far above normal levels. Preclinical investigations have demonstrated the ability of AR-RIPTAC to successfully target and kill prostate cancer cells [33]. The company Halda has assayed the ability of RIPTACs to form ternary complexes with an undisclosed EP and AR in VCaP cells, and selective apoptosis in AR-high cells was observed. This indicates that RIPTACs will indeed be able to directly target AR-V7, which is expressed in VCaP. Additionally, in VCaP xenograft mouse models, tumor growth inhibition and tumor regression were demonstrated [34]. Future studies may reveal interesting information about this drug as it transitions to IND filing, which is slated for 2024 [33].

**Table 1 cells-13-00104-t001:** Compounds that Directly Target AR-Vs. The compounds are listed along with the prostate cancer models used in preclinical studies, AR-V7 target genes tested to assess inhibitors’ activities, their mechanism of action, NCT numbers, and associated references. AR-V7 target genes tested include both the actual gene expressions and the promoters of indicated genes conjugated to a luciferase reporter. N/A: the expression of AR-V7 target genes or reporter assays using gene promoters were not tested in the cited references. *: The cited references were conference abstracts in which experimental details were not provided. This table was adapted and expanded from a section of Table 3 in the Kanayama et al. review [3].

Therapy	Prostate Cancer Models Used in Preclinical Studies	AR-V7 Target Genes Tested	Mechanism	Compound Name, Clinical Trial Number, Reference
Niclosamide	LNCaP, VCaP, CWR22Rv1, PC3, C4-2, C4-2B, xenograft (CWR22Rv1) [16]	*KLK3* (PSA) [16]	Inhibits AR-V activity via protein degradation. Reformulated niclosamide has higher bioavailability.	Niclosamide: NCT02532114 [17] Reformulated niclosamide: NCT02807805 [18]
TAS3681	DU145, xenograft (cell line not disclosed) [21]	* N/A	Downregulates AR-V7 via an unknown mechanism (the paper yet to be published).	TAS3681: NCT02566772 [22]
EPI-Compounds	LNCaP, 22RV1, MDA PCa2b, PC3, Du145, VCaP, xenograft (PC3, LNCaP, VCaP) [24,25]	*KLK3* (PSA), *TMPRSS2* [24,25] *Probasin*, *NKX3.1, UBE2C*, *AKT1, CDC20*, and *CYCLINA2* [25]	Inhibits protein-protein interaction of AR-FL and AR-Vs by targeting the AF-1 region of AR-NTD. EPI-7386 has improved bioavailability over EPI-506.	EPI-506: NCT02606123 EPI-7386: NCT04421222, NCT05075577 [27]
PROTAC	LNCaP, C4-2, CWR22Rv1, and DU145, xenograft (C4-2 and CWR22Rv1) [30]	N/A	Targets the DNA binding domain and MDM2-dependent mechanism. Clinical investigations have yet to be initiated.	Au-AR pep-PROTAC [30]
VNPP433-3β	LNCaP, CWR22RV1, xenograft (CWR22RV1) [32]	N/A	Promotes protein degradation of AR-FL and AR-Vs while also inhibiting MNK1/2. This is a galeterone drug derivative not yet in clinical trials.	VNPP433-3β [32]
RIPTAC	VCaP, xenograft (VCaP) [34]	* N/A	Selectively binds AR-FL and AR-Vs as well as an essential protein to cell viability which effectively kills cells overproducing AR.	RIPTAC [34]

## 4. Drugs That May Overcome Resistance Mediated by AR-Vs through Indirect Mechanisms

Table 2 summarizes agents that may indirectly target AR-Vs, including BET inhibitors, CBP/p300 inhibitors, PLK1 inhibitors, Bipolar Androgen Therapy (BAT), Immune Checkpoint Inhibitors, and 177Lu-PSMA-617 TAS3681. We provide a detailed review of these agents and treatments and summarize their putative mechanisms of action in Figure 4.

### 4.1. BET Inhibitors

Multiple therapies currently in clinical use or clinical trials indirectly target AR-Vs. One way this is accomplished is by targeting something downstream of AR signaling. Bromodomain and extra-terminal (BET) proteins (BRD2, BRD3, and BRD4) play a multifaceted role in cancer biology, as they can exert their influence through several mechanisms. In prostate cancer, BRD4 is most extensively studied, and one of the BRD4-mediated oncogenic mechanisms involves the elevation of key oncogenic drivers like c-Myc, which can substantially contribute to the uncontrolled growth of cancer cells. Additionally, BET proteins can enhance the activities of crucial transcription factors, including AR, AR-V7, and E2F1, through direct interaction [35]. Also, BRD4 is a member of the DNA repair complex in the non-homologous end-joining (NHEJ) pathway, and this function of BRD4 in NHEJ has been reported to mediate TMPRSS2-ERG gene fusion [36]. With this knowledge, the research community has taken a strong interest in the therapeutic potential of BET inhibitors (BETi) in the treatment of CRPC. These inhibitors target the amino-terminal bromodomains of BRD4 that typically interact with AR-FL and AR-Vs as a necessary step for transcriptional activation of AR target genes. By inhibiting this interaction, BETi epigenetically hinders prostate cancer cell growth. Several BETi are currently under development, including JQ1, ABBV-075, ODM-207, ZEN3694, PFI-1, OTX-015, NEO2734, GS-5289, and GSK525762. Most of the listed compounds have demonstrated promising anti-proliferative effects in preclinical studies [37,38,39,40,41,42,43]. 

Of note, ZEN-3694 and GS-5829 have entered clinical trial phases. A Phase Ib/IIa study evaluated the safety and effectiveness of ZEN-3694, a BETi, in combination with enzalutamide in patients with mCRPC who had shown resistance to other androgen signaling inhibitors (NCT02711956). The combination treatment showed acceptable tolerability and potential efficacy [37]. ZEN-3694 is currently in multiple phase II clinical trials (NCT04986423; NCT04471974). 

Recently, a Phase Ib/II study was conducted to investigate GS-5829 as a monotherapy and in combination with enzalutamide in mCRPC patients (NCT02607228). The compound exhibited acceptable tolerability, but its effectiveness was modest, and there was no clear correlation between dose and plasma concentration increases in individuals with mCRPC. For these reasons, the clinical development of GS-5829 has been terminated [44].

### 4.2. CBP/p300 Inhibitors

The histone acetyltransferase paralogues CREB-binding protein (CBP) and p300 are coactivators of ARs that epigenetically increase the chromatin accessibility of AR target genes. Targeted inhibition of CBP/p300 has been investigated, and a number of small molecules are being developed, including SGC-CBP30 and C646. In an investigation into mechanisms of resistance to abiraterone, SGC-CBP30 and C646 demonstrated synergistic effects with abiraterone, as they mitigated the phosphorylation of CREB1 and subsequent enhanced activity of the CBP/p300 complex [45]. A recent preclinical investigation into a novel small-molecule CBP/p300 inhibitor, FT-6876, resulted in anti-proliferative activity in AR-positive breast cancer models in vitro and in vivo [46]. The efficacy of this compound could translate to prostate cancer models as well, so further pursuit of this research is necessary.

A novel small-molecule inhibitor of the CBP/p300 bromodomain, CCS1477, has been validated as a potential therapeutic strategy for lethal prostate cancer. It has shown efficacy in inhibiting cell proliferation, reducing AR and c-Myc-regulated gene expression, and exhibiting anti-tumor activity in AR-V-driven models such as CWR22Rv1, VCaP, and LNCaP 95. Early clinical studies have indicated that CCS1477 can modulate KLK3 blood levels and regulate CRPC biopsy biomarker expression [47]. Two Phase I/IIa studies on CCS1477 are ongoing. One is investigating the effectiveness of CCS1477 as a monotherapy and in combination with enzalutamide or abiraterone (NCT03568656). The other is investigating its application to other forms of cancer (NCT04068597).

As the molecular components of cancer biology are a complexly interwoven narrative, CBP/p300 can be targeted by a novel dual inhibitor of BET and CBP/p300, NEO2734. NEO2734 has been shown to be an effective therapeutic intervention for *SPOP*-mutant prostate cancer in preclinical studies. *SPOP* is the most frequently mutated gene in primary prostate cancer, and its loss of function causes excessive accumulation of its substrates and deregulation. Wild-type *SPOP* typically binds to and induces ubiquitination and proteasomal degradation of BET proteins. Patients with *SPOP* mutations experience BETi resistance. NEO2734 efficiently inhibited *SPOP*-mutated prostate cancer cell growth in preclinical investigations [43]. An additional study found that enzalutamide-resistant prostate cancer cells, organoids, and PDXs were susceptible to NEO2734, suggesting a potential treatment strategy [48]. Recent investigations into AR-V mechanisms of increasing castration resistance utilizing NEO2734 have also demonstrated that they counteract anti-androgen-induced ferroptosis. NEO2734 was a successful inhibitor of this process [49]. There is currently an ongoing Phase I dose-escalation study investigating NEO2734 as a monotherapy in patients with CRPC (NCT05488548).

### 4.3. PLK1 Inhibitors

Another interesting drug target that has indirect effects on the function of AR-Vs is Polo-like kinase 1 (PLK1). PLK1 has been extensively investigated, and it collaborates with other kinases, such as cyclin-dependent kinase 1 (CDK1) and Aurora A or Aurora B, to control critical cell cycle events. PLK1 inhibitors have shown promise as potential cancer drugs due to their role in regulating mitotic events and the proliferative activity of cancer cells. However, the selectivity of these inhibitors is crucial, as other PLK family members like PLK2, PLK3, and PLK4 may have different roles in cancer. PLK1’s unique structure, including its polo-box domain, makes it an attractive target for drug development [50]. PLK1 is elevated in prostate cancer and plays a role in coordinating the downstream PI3K-AKT-mTOR pathway and the activation of AR signaling through elevating cholesterol biosynthesis during the progression to castration-resistant disease [51]. In the context of AR-V7, a few potential mechanisms through which a PLK1 inhibitor affects AR-V7 have been proposed. Zhang et al. showed that PLK1 inhibition resulted in the synergetic reduction in AR-V7 protein in combination with abiraterone [51]. By contrast, Patterson et al.’s findings suggested that this synergistic effect is brought on by abiraterone’s AR-independent effect on cell mitosis. Briefly, abiraterone as monotherapy causes an increase in the percentage of mitotic cells, and when combined with a PLK1 inhibitor, two drugs induce a two-fold increase in mitotic arrest compared to onvansertib monotherapy [52]. In both studies, PLK1, in combination with abiraterone, suppressed the growth of AR-V7-expressing cells [51,52]. Since the exact inhibitory mechanism of the PLK1 inhibitor towards AR-V7 is yet to be fully clarified, we categorized this drug as an “indirect drug”. Currently, a Phase II clinical trial of onvansertib, a PLK1 inhibitor, is being conducted to evaluate its effectiveness in combination with abiraterone and prednisone (NCT03414034).

### 4.4. Bipolar Androgen Therapy

A recently popularized treatment that challenges prior conceptions is bipolar androgen therapy (BAT). In HSPC prostate cancer cells, AR acts as a licensing factor for DNA replication, requiring degradation during each cell cycle to enable DNA replication in the subsequent cycle. Experiments using HSPC cell lines and in vivo xenograft models confirm that AR levels fluctuate throughout the cell cycle and are specifically degraded during mitosis, supporting the role of AR as a licensing factor for DNA replication in HSPC cells. In contrast, CRPC cells do not exhibit AR degradation during mitosis [53]. Additionally, androgen has a dose-dependent biphasic effect on cell proliferation [54]. This significant regulation and sensitivity of androgen renders CRPC cells vulnerable to not only sub-physiological exogenous androgen but also to supraphysiological exogenous androgen. This is the inspiration behind BAT, which exploits this vulnerability in CRPC cells [55]. Paradoxically, high androgen levels can inhibit prostate cancer growth by inducing DNA double-strand breaks and cell cycle arrest, particularly in tumors with DNA repair deficiencies, suggesting potential therapeutic strategies involving androgen treatment combined with PARP inhibition for certain prostate cancer cases [56]. 

BAT is comprised of cyclic treatment with testosterone supplementation followed by a washout period that rapidly alters serum androgen levels between two extremes. The TRANSFORMER study evaluated BAT in comparison to enzalutamide for the treatment of CRPC (NCT02286921). The study found that BAT showed meaningful clinical activity and safety, with PFS similar to enzalutamide. BAT resulted in a higher PSA-PFS when patients were crossed over to enzalutamide. The study suggests that BAT can sensitize CRPC to subsequent anti-androgen therapy [57]. 

In another study that compared BAT in post-enzalutamide and post-abiraterone cohorts, BAT showed clinical activity in mCRPC patients, with a higher response rate when challenged with enzalutamide compared to abiraterone (NCT02090114). Patients with detectable AR-V7 in circulating tumor cells had worse outcomes. BAT followed by AR-targeted therapy rechallenge did not improve outcomes in AR-V7-positive patients [58]. Perhaps BAT in combination therapy will yield synergistic effects that improve outcomes for AR-V7-positive patients. Immune checkpoint blockade may have therapeutic potential in prostate cancer patients, especially following treatment with BAT and enzalutamide [59]. There are currently several ongoing Phase II clinical trials investigating BAT in combination with other therapies such as nivolumab (NCT03554317) [60], radium 223 (NCT04704505), darolutamide (NCT04558866), carboplatin (NCT03522064), and olaparib (NCT03516812).

### 4.5. Immune Checkpoint Inhibitors

Immune checkpoint inhibitors, such as nivolumab and ipilimumab, work by targeting specific molecules involved in regulating the immune response, allowing the immune system to better recognize and attack cancer cells. Nivolumab is a monoclonal antibody that targets the programmed cell death protein 1 (PD-1) receptor on immune cells. PD-1 is a checkpoint receptor that, when bound to its ligands PD-L1 and PD-L2 on cancer cells, suppresses the immune response and prevents immune cells from attacking the cancer. Nivolumab blocks the interaction between PD-1 and its ligands, thereby “releasing the brakes” on the immune system and enabling it to attack cancer cells more effectively. Ipilimumab, on the other hand, targets a different immune checkpoint called cytotoxic T-lymphocyte antigen 4 (CTLA-4). CTLA-4 is another receptor on immune cells that downregulates the immune response. Ipilimumab inhibits CTLA-4, preventing its inhibitory signal and allowing immune cells to remain active and attack cancer cells [61]. 

Immune checkpoint inhibitors are an additional therapy that potentially indirectly targets AR-Vs. Although prostate cancer is typically considered to have a low tumor mutational burden, AR-V7-positive prostate cancers have been associated with a greater number of DNA-repair gene mutations and a higher tumor mutational burden [62]. Theoretically, this makes AR-V7-positive patients potentially better candidates for therapy via immune checkpoint inhibitors than the general prostate cancer patient population. 

Nivolumab and ipilimumab are often used in combination to result in a more potent and synergistic immune response against cancer cells. A Phase II clinical trial testing ipilimumab plus nivolumab in patients with AR-V7-positive advanced prostate cancer (NCT02601014) showed satisfactory safety and promising effectiveness in individuals with advanced prostate cancer expressing AR-V7, provided they also carry DNA-repair abnormalities. This efficacy was not consistent in all AR-V7-positive patients [63]. An additional cohort of the Phase II trial investigated nivolumab plus ipilimumab with or without enzalutamide in AR-V7-positive mCRPC patients. The findings indicate that the use of nivolumab along with ipilimumab at these doses shows acceptable safety, although it yields only moderate effectiveness in individuals with AR-V7-positive prostate cancer, even in combination with enzalutamide [64]. 

The CheckMate 650 Phase II trial expanded the study population to include CRPC patients regardless of AR-V7 status to gain better insight into the comparative benefits of nivolumab and ipilimumab vs. traditional chemotherapy with cabazitaxel (NCT02985957). Preliminary results suggested that immune checkpoint inhibitors are an effective therapy for a subset of CRPC patients, with four participants experiencing complete responses [65], but there was no clear and consistent association between efficacy and tissue or blood tumor mutational burden in the nivolumab plus ipilimumab cohorts [66]. The final results should yield interesting conclusions, as this study included 351 participants when it closed to enrollment. Further research is necessary to determine whether these therapies should be utilized in AR-V7-positive patients with CRPC. There are currently 13 active trials investigating nivolumab and ipilimumab in combination with additional therapies such as ^177^Lu-PSMA (NCT05150236) and SBRT (NCT05655715).

A relatively new and promising alternative immunotherapy that has been developed for metastatic prostate cancer treatment is pTVG-AR (MVI-118). This immunotherapy is a DNA vaccine encoding the LBD of AR that consequently induces a CD8+T cell-mediated immune response against cancer cells overexpressing AR. In a Phase I trial (NCT02411786), the utilization of pTVG-AR demonstrated safety and activation of the immune system in individuals diagnosed with mHSPC. The correlation observed between immune response and PSA-PFS implies that this treatment approach could potentially extend the period before castration resistance develops, aligning with findings from preclinical studies [67]. There are currently two ongoing Phase II clinical trials that include the pTVG-AR DNA vaccine. One centers on the pTVG-HP DNA vaccine, with or without the pTVG-AR DNA vaccine and pembrolizumab, in patients with mCRPC (NCT04090528). The other investigates ADT, with or without pTVG-AR, and with or without nivolumab, in patients with newly diagnosed, high-risk prostate cancer (NCT04989946). While the Phase I study showed promising outcomes, it will be important to assess the efficacy of this therapy with respect to AR-V7-positive CRPC, which lacks the LBD that the vaccine is preparing the immune system against.

### 4.6. ^177^Lu-PSMA-617 (Pluvicto)

Another therapy targeting AR-V-expressing cells that simultaneously express prostate-specific membrane antigen (PSMA) is ^177^Lu-PSMA-617, otherwise known as pluvicto. Newly approved by the U.S. Food and Drug Administration, this drug is comprised of a PSMA-targeting molecule linked to the radioactive isotope lutetium-177 (^177^Lu). The therapy has shown incredibly promising results in its Phase III VISION trial (NCT03511664) [68] for the treatment of late-stage PSMA-positive CRPC, and combination therapies with other drugs are being tested in clinical trials (NCT05340374 [69], NCT05113537). The PSMA targeting molecule allows the therapy to specifically bind to PSMA-expressing prostate cancer cells, delivering radiation directly to the tumor cells while minimizing damage to surrounding healthy tissues. The radiation emitted from ^177^Lu damages the DNA within the cancer cells, leading to their destruction [70]. This therapy is currently only approved for use in individuals with mCRPC who have exhausted all available treatment options. 

A recent study investigated the molecular biomarkers in the circulating tumor cells of patients undergoing treatment with Pluvicto. The study analyzed clinical and molecular parameters, including PSA and PSMA mRNA expression, tumor volumes, and gene expression of AR-FL and AR-V7. The study found that AR-FL and AR-V7 may serve as prognostic biomarkers for high tumor burden in mCRPC patients prior to treatment with Pluvicto, but none of these parameters correlated with response to PSMA treatment. Of note, AR-V7-positive status did correlate with higher levels of AR, a higher tumor load, and increased PSMA expression in patients [71]. Interestingly, common neuroendocrine differentiation markers were associated with longer overall survival but not PFS. Similarly, Pathmanandavel et al. [72] reported findings where 11 AR-V7-positive patients were treated by ^177^Lu-PSMA-617, and the presence of AR-V7 did not negatively affect treatment response or survival, suggesting that ^177^Lu could be another primary treatment option for AR-V7-positive mCRPC in addition to taxanes. Further investigation is needed to analyze the potential role of protein expression in these markers. There are numerous active clinical trials involving ^177^Lu-PSMA-617. Some of these include the PSMAfore trial (NCT04689828), which compares ^177^Lu-PSMA-617 against AR-targeted therapy in mCRPC patients who are taxane-naïve; a combination trial of Pluvicto and pembrolizumab (NCT03805594); and a study investigating mechanisms of resistance to PSMA radioligand therapy (NCT05435495). As further results are published from a number of ongoing studies, we may gain further insight into the efficacy of treating AR-V7-positive patients with ^177^Lu-PSMA-617.

**Table 2 cells-13-00104-t002:** Compounds that Indirectly Target AR-Vs. This table lists compounds that indirectly target AR-Vs, along with the prostate cancer models used in preclinical studies, their mechanisms of action, NCT numbers, and associated references. The table was adapted and expanded from a section of Table 3 in the Kanayama et al. review [3].

Therapy	Prostate Cancer Models Used in Preclinical Studies	Mechanism	Compound Name, Clinical Trial Number, Reference
Bromodomain and Extra Terminal (BET) Inhibitors	LNCaP, CWR22Rv1, LNCaP95 [36]	Targets the amino terminal bromodomains of BRD4, which typically interact with AR-FL and AR-Vs as a necessary step for transcription activation of AR target genes. BRD4 also elevates c-Myc, interacts with E2F, and mitigate TMPRSS2-ERG fusion.	ZEN-3694: NCT02711956 [37] NCT04986423 NCT04471974 GS-5829: NCT02607228 [44]
CREB-binding Protein (MCBP)/p300 Inhibitors	LNCaP, PC-3, LNCaP-abl, LAPC-4, CWR22Rv1, VCaP, LNCaP-AR, LNCaP-Bic, C4-2, LNCaP95, DU145 [45,47], xenograft (CWR22Rv1), PDX (CP-50) [47]	Targeted inhibition of CBP/p300 enzymes decreases chromatin accessibility and lowers the transcription of AR target genes. NEO2734 is a dual CBP/p300 and BET Inhibitor	CCS1477: NCT03568656 [47] NEO2734: NCT05488548 [49]
Polo-like Kinase 1 (PLK1) Inhibitors	RWPE-1 LNCaP, C4-2, CWR22Rv1, MR49F, LNCaP95, PC3, DU145 [51,52], PDX (LuCaP35CR) [51], PDX (LVCaP-2CR) [52]	Target inhibition of PLK1 in combination with abiraterone results in reduced AR-FL and AR-V7 protein expression potentially via cholesterol biosynthesis inhibition. Also, the drug combination induces mitotic arrest.	Ovansertib: NCT03414034
Bipolar Androgen Therapy (BAT)	LNCaP, PC-3, CWR22Rv1, PrSC, LAPC-4 [53], VcaP, DU145 [56], xenograft (CWR22Rv1, LNCaP, LAPC-4) [53], LNCaP 104-S, LNCaP 104-R2, xenograft (LNCaP 104-S, 104-R1, CDXR-3) [55], Ex vivo PDX culture (LuCaP 35, 70, and 96CR) [56]	The cyclic alternation between two extremes of blood serum testosterone levels re-sensitizes CRPC to anti-androgen therapies. BAT in combination therapy may yield synergistic effects that improve outcomes for AR-V7 positive patients.	BAT: NCT02286921 [57] NCT02090114 [58] NCT04704505 NCT04558866 NCT03522064 NCT03516812
Immune Checkpoint Inhibitors	Cited papers did not include preclinical studies.	Nivolumab and ipilimumab target specific molecules (PD-1 and CTLA-4 respectively) involved in regulating the immune response, allowing the immune system to better recognize and attack cancer cells. pTVG-AR (MVI-118) is a DNA vaccine encoding the ligand binding domain of AR, which induces CD8+T cell-mediated immune response against cancer cells overexpressing AR.	Nivolumab + ipilimumab: NCT02601014 [63] NCT02985957 [65] NCT05150236 NCT05655715 PTVG-AR (MVI-118): NCT02411786 [67] NCT04090528 NCT04989946 NCT03554317
^177^Lu-PSMA-617 (Pluvicto)	Cited papers did not include preclinical studies.	PSMA ligand linked to the radioactive isotope, ^177^Lu, targets PSMA-expressing cells.	NCT03511664 NCT05340374 NCT05113537

## 5. Conclusions

The intricate landscape of prostate cancer progression, marked by the emergence of AR-Vs, has prompted an intensive exploration of novel therapeutic approaches to tackle the challenges posed by these variants. As prostate cancer research continues to uncover the crucial roles of AR-Vs in treatment resistance and disease progression, the urgency to develop effective interventions becomes evident. The therapeutic landscape is evolving rapidly, with promising therapies like reformulated niclosamide, EPI compounds, PROTAC molecules, VNPP433-3β, TAS3681, and RIPTACS demonstrating their ability to directly target AR-Vs. Compounds that indirectly target AR-Vs hold the potential to modulate upstream pathways and molecular processes that influence AR-V expression and activity, offering a multifaceted approach to mitigating their impact on prostate cancer progression. Given that the generation of AR-Vs is prostate cancer’s evolutionary adaptive response to ADT, it is strategic to attack prostate cancer’s Achilles’ heel from different angles with indirect targeting agents. Since long-term intensive inhibition of AR signaling renders prostate cancer lethal and AR-indifferent lineages, it is crucial to eliminate them before this transformation occurs. 

Comprehensively understanding the mechanisms of action of these compounds and their potential clinical impact can pave the way toward personalized and precise treatments that could revolutionize the care of advanced prostate cancer patients. 

## 6. Future Directions

Ongoing clinical trials, including those evaluating combination therapies, hold the promise of reshaping treatment paradigms, potentially leading to improved patient outcomes. Ultimately, the pursuit of therapeutic strategies aimed at AR-Vs signifies a pivotal step forward in addressing therapeutic resistance and advancing the precise management of advanced prostate cancer.

## Figures and Tables

**Figure 2 cells-13-00104-f002:**
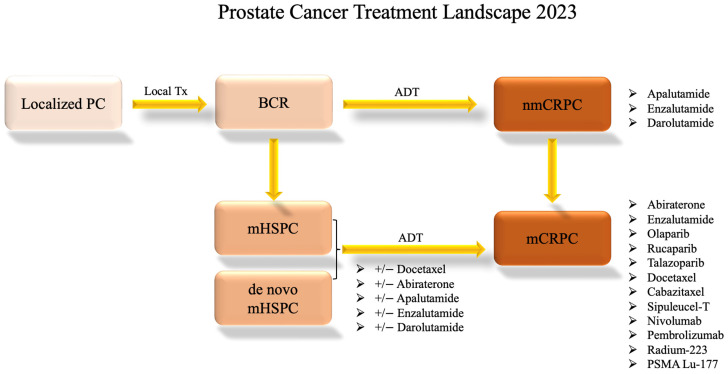
Prostate Cancer Systemic Treatment Landscape. This flow chart shows the standard progression through available treatment options for varying disease statuses as of 2023. Clinical trial options are not included. The information presented is subject to change pending future drug approvals. ADT: androgen-deprivation therapy; BCR: biochemical recurrence; local Tx: local treatment such as radiation or surgery; mCRPC: metastatic castration-resistant prostate cancer; mHSPC: metastatic hormone-sensitive prostate cancer; nmCRPC: non-metastatic castration-resistant prostate cancer; PC: prostate cancer.

**Figure 3 cells-13-00104-f003:**
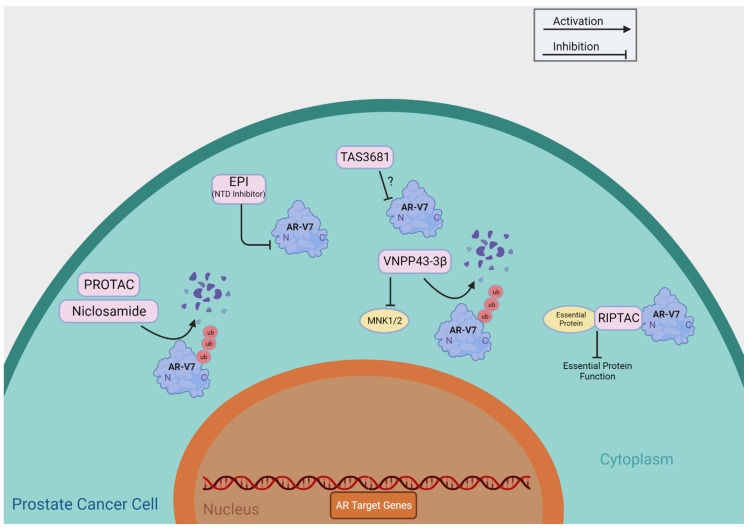
Direct Targeting of AR-V7. This figure provides a brief overview of the mechanisms of action of the compounds that directly target AR-V7, as shown in Table 1. ?: The mechanism of action of TAS3681 is yet to be published.

**Figure 4 cells-13-00104-f004:**
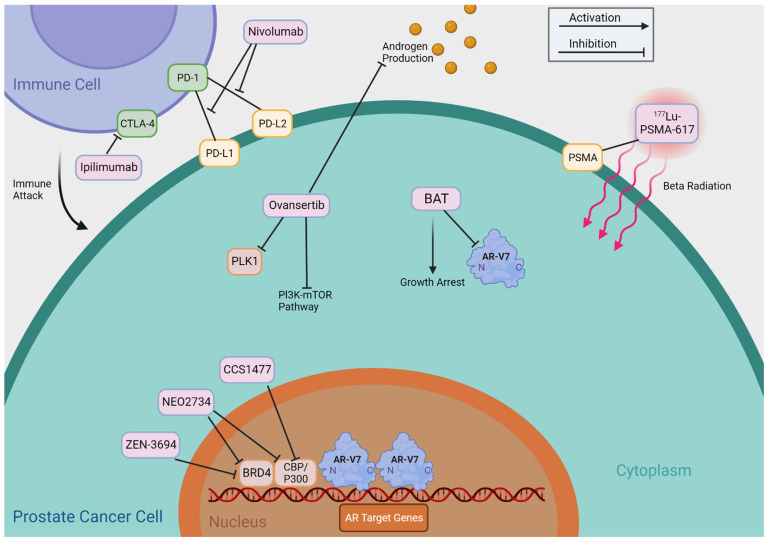
Mechanisms of Indirectly Targeting AR-V7. This figure provides an abbreviated overview of the mechanisms of action of the therapies that indirectly target AR-Vs, as shown in Table 2.

## Data Availability

Not applicable.

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
