# Peer review of "Therapeutic Approaches to Targeting Androgen Receptor Splice Variants"

_cells, 2024, doi:10.3390/cells13010104_

Round 1

Reviewer 1 Report

Comments and Suggestions for Authors

The paper entitled: “Therapeutic Approaches to Targeting Androgen Receptor 2

Splice Variants” authored by Daniels et al., is designed to provide insight into the therapies that target AR splice variants in cancer therapy. The authors present a summary of the current data and clinical trials to date, which is both important and timely. This manuscript is well written and provides great insight into fundamental information into the current and future implications of targeting ARs directly and indirectly and how they may be used ultimately for translation into the clinic. This is easy to read, brief, and should be of interest to many of the biomedical community that study prostate cancer and castration resistance. Only a few minor concerns:

1.     Line 47, “…truncated isoforms of the AR that no longer bind the androgens but have the potential…”, 

This sounds like the AR variant used to bind androgens and no longer does. Please clarify/reword.

2.      Lines 127-128 “Niclosamide's mechanism of ac-tion for prostate cancer involves hindering AR-V7 recruitment to the PSA promoter and …”.  I presume this was an observation of what happened, rather than mechanism of action.  I would expect the drug to prevent ARv7 from binding to a number of different AR/ARE mediated genes including PSA.  Please clarify/reword

Author Response

Response to reviewers:

We sincerely thank the reviewers for their time and their positive comments. We have revised the original manuscript to address comments raised by reviewers. The changes in the manuscript are highlighted in yellow. 

Reviewer 1

The paper entitled: “Therapeutic Approaches to Targeting Androgen Receptor Splice Variants” authored by Daniels et al., is designed to provide insight into the therapies that target AR splice variants in cancer therapy. The authors present a summary of the current data and clinical trials to date, which is both important and timely. This manuscript is well written and provides great insight into fundamental information into the current and future implications of targeting ARs directly and indirectly and how they may be used ultimately for translation into the clinic. This is easy to read, brief, and should be of interest to many of the biomedical community that study prostate cancer and castration resistance. Only a few minor concerns:

  1. Line 47, “…truncated isoforms of the AR that no longer bind the androgens but have the potential…”, This sounds like the AR variant used to bind androgens and no longer does. Please clarify/reword.

We appreciate the reviewer’s due diligence. It was rephrased to “AR-Vs are truncated isoforms of the AR that do not bind the androgens”.

  1. Lines 127-128 “Niclosamide's mechanism of action for prostate cancer involves hindering AR-V7 recruitment to the PSA promoter and …”.  I presume this was an observation of what happened, rather than mechanism of action.  I would expect the drug to prevent ARv7 from binding to a number of different AR/ARE mediated genes including PSA.  Please clarify/reword.

We agree with the reviewer that decreased AR-V7 recruitment to the PSA promoter is an observation that led the authors to conclude that niclosamide hampers the transcription activity of AR-V7. The cited Liu et al. papers only showed decreased AR-V7 binding to the PSA promoter by ChIP assay, and comprehensive ChIP sequencing has not been performed. Thus, we refrained from definitively stating that Niclosamide prevents AR-V7 from binding to several AREs, and rewrote the sentence as follows: Niclosamide's mechanism of action for prostate cancer is expected to involve hindering AR-V7 recruitment to androgen-responsive elements such as the PSA promoter….”.

Reviewer 2 Report

Comments and Suggestions for Authors

Well-organized. Figures can have more details. It would be informative to have a section on future directions.

Figure 1; Please add amino acid numbers to define the boundaries of each domain within the schematic.

Line 161: Please provide more details on TAS3681 ability to inhibit AR movement. What is the putative mechanism of action. If mechanism is not known, how was it developed? Any knowledge on which domain it is binding to within AR.

In Table I, please add a column showing which pre-clinical PC models have been tested for the agents prior to clinical trials. Please include the appropriate references.

Expand RIPTAC abbreviation on first use.

In Figure 3 or in a table format please add details of which target genes were tested to measure the activity of each AR-V7 inhibitor.

Line 289, which AR-v models; please specify.

What are the other major targets of BET inhibitors in mCRPC besides MYC, AR and AR-V7.

How do PLK1 inhibitors interfere with the function of AR-V7.

Please provide an indepth description of the model tested in pre-clinical testing for the inhibitor.

Many references are incompletely listed. Volume numbers are missing.

For e.g. reference 10, 20-journal, volume and page numbers are missing.

references 32, 57, 59, 63, 66; journal information is missing.

Please check all references carefully and thoroughly.

Author Response

Response to reviewers:

We sincerely thank the reviewers for their time and their positive comments. We have revised the original manuscript to address comments raised by reviewers. The changes in the manuscript are highlighted in yellow.

Reviewer 2 

  1. Well-organized. Figures can have more details. It would be informative to have a section on future directions.

We appreciate your comments. We elaborated on the conclusion and moved the latter part of the conclusion to future directions in the end.

  1. Figure 1; Please add amino acid numbers to define the boundaries of each domain within the schematic.

Amino acid numbers were added to Figure 1.

  1. Line 161: Please provide more details on TAS3681 ability to inhibit AR movement. What is the putative mechanism of action. If mechanism is not known, how was it developed? Any knowledge on which domain it is binding to within AR.

We agree with a reviewer that TAS3681’s mechanism of action is not fully explained in this manuscript. Initially, the drug was developed by Taiho Pharmaceutical Company, and because it was not developed by an academic institution, the paper about the development of this drug has not been published yet (can be related to patents). This is why all cited references regarding this drug are poster abstracts, and the detailed mechanism was not disclosed in those abstracts. Hopefully, it will soon be published.

  1. In Table I, please add a column showing which pre-clinical PC models have been tested for the agents prior to clinical trials. Please include the appropriate references.

We agree with a reviewer that it is informative. We added the column of preclinical models with references in Table 1.

  1. Expand RIPTAC abbreviation on first use.

The full name of RIPTAC was added on the first use.

  1. In Figure 3 or in a table format please add details of which target genes were tested to measure the activity of each AR-V7 inhibitor.

A column was added. Some papers did not test AR-V7 target genes, but instead focused on AR-V7 protein degradation itself. In that case, it is indicated as N/A. Additionally, for newly developed drugs such as TAS3681 and RIPTAC, for which articles are yet to be available, we had to cite the conference poster abstracts in which experimental details are not provided. Thus, for these drugs, it is possible that AR-V7 target genes were tested, but not described in the abstract. The footnote was added to the legends, accordingly.

  1. Line 289, which AR-v models; please specify.

They were CWR22Rv1, VCaP and LNCaP. They were added to the manuscript.

  1. What are the other major targets of BET inhibitors in mCRPC besides MYC, AR and AR-V7.

BRD4 physically interacts with E2F1, and is also involved in the non-homologous end-joining pathway, which subsequently mediates TMPRSS2-ERG gene fusion. This has been added to the manuscript.

  1. How do PLK1 inhibitors interfere with the function of AR-V7.

We agree with a reviewer that the description of PLK1 inhibitor’s interaction with AR-V7 was missing. PLK1inhibitor, in combination with abiraterone, synergistically inhibits the growth of AR-V7-expressing cells. Two potential mechanisms have been proposed so far: AR-V7 protein reduction and increased mitotic cell arrest. We elaborated on the mechanism and added Patterson et al.’s reference.

  1. Please provide an in-depth description of the model tested in pre-clinical testing for the inhibitor.

 We added the column of preclinical models with references in Table 2.

  1. Many references are incompletely listed. Volume numbers are missing.

We updated the reference in Endnote and added DOI when available. The cited poster abstracts were missing journal names and page numbers, so they were manually added.

  1. For e.g. reference 10, 20-journal, volume and page numbers are missing.

For reference 10 (Luo, J. Development of AR-V7 as a putative treatment selection marker for metastatic castration-resistant prostate cancer. Asian journal of andrology), volume was 18 and page is 580. It was added to the reference. For Reference 20 (Seki, M. TAS3681, a novel type of AR antagonist with AR downregulating activity, as a new targeted therapy for aberrant AR-driven prostate cancer), journal name, volume and page were added.

  1. Please check all references carefully and thoroughly.

We appreciate the reviewer’s due diligence. The cited poster abstracts published in the Journal of Clinical Oncology were missing journal names, volume, and page information. The information was added.

Round 2

Reviewer 2 Report

Comments and Suggestions for Authors

Line 340: Please correct typo "indirect dug" to "indirect drug".

Line 347: Please revise this  sentence: "The final therapy that indirectly targets AR-Vs is 177Lu-PSMA-617.....". It is a little confusing. Is AR-V the target? or is it AR-V expressing cells?  If, so it may be revised, "to another therapy indirectly targeting AR-V  expressing cells, also expressing PSMA is the 177Lu-PSMA-617" RLT".

Author Response

Response to reviewers:

We sincerely thank the reviewers for spending time to improve our manuscript. We corrected the manuscript accordingly. The changes in the manuscript are highlighted in yellow.

Reviewer 2

  • Line 340: Please correct typo "indirect dug" to "indirect drug".

 Thank you very much for catching the typo. It was corrected.

  • Line 437: Please revise this  sentence: "The final therapy that indirectly targets AR-Vs is 177Lu-PSMA-617.....". It is a little confusing. Is AR-V the target? or is it AR-V expressing cells?  If, so it may be revised, "to another therapy indirectly targeting AR-V  expressing cells, also expressing PSMA is the 177Lu-PSMA-617" RLT".

We appreciate a reviewer’s suggestion. The target is AR-V-expressing cells. The sentence was corrected accordingly.